# Anatomy Nights: An international public engagement event increases audience knowledge of brain anatomy

Katherine A. Sanders[1◉*], Janet A. C. Philp[2◉], Crispin Y. Jordan[2], Andrew S. Cale[3], Claire L. Cunningham[4], Jason M. Organ[3,5]

1 Centre for Anatomical and Human Sciences, Hull York Medical School, University of Hull, Hull, United Kingdom, 2 Deanery of Biomedical Sciences, Edinburgh Medical School, University of Edinburgh, Edinburgh, United Kingdom, 3 Department of Anatomy, Cell Biology & Physiology, Indiana University School of Medicine, Indianapolis, Indiana, United States of America, 4 Centre for Anatomy and Human Identification, Medical Sciences Institute, University of Dundee, Dundee, United Kingdom, 5 Department of Communication Studies, Indiana University Purdue University Indianapolis, Indianapolis, Indiana, United States of America

◉ These authors contributed equally to this work.
* Kat.Sanders@hyms.ac.uk

**Data Availability Statement:** All relevant data are within the manuscript and its Supporting Information files. We have clearly described all of the statistical tests used as well as which functions

## Abstract

Anatomy Nights is an international public engagement event created to bring anatomy and anatomists back to public spaces with the goal of increasing the public's understanding of their own anatomy by comparison with non-human tissues. The event consists of a 30-minute mini-lecture on the anatomy of a specific anatomical organ followed by a dissection of animal tissues to demonstrate the same organ anatomy. Before and after the lecture and dissection, participants complete research surveys designed to assess prior knowledge and knowledge gained as a result of participation in the event, respectively. This study reports the results of Anatomy Nights brain events held at four different venues in the UK and USA in 2018 and 2019. Two general questions were asked of the data: 1) Do participant post-event test scores differ from pre-event scores; and 2) Are there differences in participant scores based on location, educational background, and career. We addressed these questions using a combination of generalized linear models (R's glm function; R version 4.1.0 [R Core Team, 2014]) that assumed a binomial distribution and implemented a logit link function, as well as likelihood estimates to compare models. Survey data from 91 participants indicate that scores improve on post-event tests compared to pre-event tests, and these results hold irrespective of location, educational background, and career. In the pre-event tests, participants performed well on naming structures with an English name (frontal lobe and brainstem), and showed signs of improvement on other anatomical names in the post-test. Despite this improvement in knowledge, we found no evidence that participation in Anatomy Nights improved participants' ability to apply this knowledge to neuroanatomical contexts (e.g., stroke).

in R were used. Furthermore, we provide supplementary files that include all the raw data from the surveys (S3), all estimates from generalized linear mixed models (S2), and the R scripts used to analyze the data (S4) so that anyone interested can re-run the analyses if they wish.

**Funding:** The authors received no specific funding for this work.

**Competing interests:** The authors have declared that no competing interests exist.

## Introduction

The public has always had a fascination with the human body. Public dissections were historically led by experts of anatomy, including Mondino De Luzzi (14th Century) [1], and Andreas Vesalius (16th Century) [2, 3]. More recently, the presentation of previously dissected human bodies by BODY WORLDS has become phenomenally popular, each adding something to the public understanding of anatomy [4, 5]. Here, we describe Anatomy Nights, a new public engagement format that returns the art of dissection to public audiences.

Anatomy Nights was created to bring anatomy and anatomists back to public spaces and audiences. It is an event series coordinated by a central team and provides the necessary tools to enable anatomists to engage with a local public audience in local public spaces [6]. Through the presentation of human anatomical concepts and demonstration of these concepts via dissection of animal tissue, Anatomy Nights' goal is to increase the public's understanding of their own anatomy by reference to non-human tissues (e.g., lamb, pig), and to link this anatomical knowledge to common health conditions such as stroke.

Anatomy has long been considered a cornerstone of medical education, and the development of public knowledge of anatomy can be considered important in promoting health literacy [7]. This is particularly significant as low health literacy is associated with poorer health outcomes [8]. Over 100,000 and 795,000 people are affected by stroke each year in the UK [9] and the USA [10], respectively, and these figures are expected to rise. Stroke is an anatomically related medical condition that affects a large proportion of our population, and yet the public's understanding of their own bodies has been demonstrated multiple times to be lacking [11, 12]. Whilst members of the public can generally correctly identify that the brain is located within the skull [11], how it works and how injuries can affect it are not so widely understood. Coupled with an overestimation of the publics' medical knowledge by the clinical professions [13, 14], this lack of understanding of their own bodies can lead to communication issues about medical diagnoses and treatment procedures leaving patients, and their families confused and anxious [15], impacting patient care.

As previously stated, part of the Anatomy Nights event includes dissection of non-human tissue to demonstrate key, homologous anatomical structures. The absence of dissection of human tissue is due to moral and legal considerations around what constitutes appropriate use of donated human tissue, and this can be a barrier to anatomists being able to take part and host their own Anatomy Nights events. These considerations are exemplified by the existence of national legislation within Anatomy Nights' host countries. In the United Kingdom (UK), the use and display of human tissue are regulated by the Human Tissue Act 2004 and the Human Tissue (Scotland) Act 2006. These acts were preceded by The Anatomy Act [16] in 1832, which allowed cadaveric specimens to be used only by approved medical schools. This restriction still exists, and it is this caveat that stops the British public from accessing education of human anatomy through viewing anatomical dissection by an expert. Similarly, in the United States of America (USA), the Uniform Anatomical Gift Act legislation was adopted by 26 of 50 states in 1968 [17], and revisions in 1987 and 2006 have resulted in 48 states adopting uniform laws related to body and organ donation [18].

Regulation of anatomical specimens for public display, combined with fewer opportunities for the public to engage around this material with qualified anatomists, has resulted in a lack of public access to anatomy. Therefore, non-academic demonstrations of anatomical preparations without proper educational context, such as the traveling plastination exhibitions, have become popular throughout Europe and the United States. The spate of non-contextualized information (or even misinformation) can be confusing, leaving the public with a misunderstanding of anatomical structure and function similar to that seen in patients who forego

medical advice in favor of internet self-diagnosis [19]. While some establishments have tried to address this [20], no data has been collected as to the effect of the courses, and the high cost of such events is a limiting factor for engagement. To minimize the impact of financial barriers, Anatomy Nights events are accessible to the public for a nominal ticket fee (around the price of a coffee in each host's country). The majority of the fee is donated to a relevant charity, and the rest used for funding the next events.

Here we present our evaluation of a brain-focused Anatomy Nights event and its impact on increasing public knowledge of anatomy. Specifically, the research aims were to determine if the Anatomy Nights events had a positive impact on the public audiences' knowledge of brain anatomy. In particular, the goals were to establish which anatomical features of the brain are common knowledge, where there is a deficit, and whether audiences could take this knowledge and apply it to neuroanatomical concepts.

## Materials and methods

### Event format

Data were collected at four different venues at events during October 2018—October 2019 (Dundee, Edinburgh, Hull, UK, and Indianapolis, USA). Each event was hosted by a different anatomist using a template presentation covering the anatomical knowledge that was tested. The venues were all public settings, specifically in venues not associated with universities to encourage attendance from anyone who was interested but may not feel comfortable in a university setting [6]. The events were advertised as a short talk about the brain followed by a dissection of a lamb or pig's brain.

Each of the Anatomy Nights events followed a standard format. This started with a 30-minute talk on the anatomy of the brain by the hosting anatomist/s, including where it is; meninges; white and grey matter; lobes and the cerebellum; functional areas and the homunculus; decussation of fibers; basic blood supply; CSF and the ventricles. Following this talk, a dissection of a non-human brain was conducted. To ensure all members of the audience could see this clearly, a camera-projector rig was set up. In the dissection, the audiences were familiarized with the external structure, including lobes and brainstem. The brain was then cut into sections to demonstrate white vs. grey matter, points of decussation, and the ventricles. Any significant deviations from human anatomy (e.g., the olfactory lobe seen on pigs brains) were highlighted to the audience.

### Data collection

Everyone in attendance aged 16 and over was invited to participate in research surveys designed to assess baseline and acquired knowledge as a result of participation in Anatomy Night. Data were collected through two separate instruments—before and after the events— which enabled assessment of the existing knowledge gap and whether the learning program was effective. For those attendees to choose to participate in the research surveys, each received a participant information sheet alongside the survey for them to retain for their records. At the top of the surveys, participants were notified that submission of the surveys at the conclusion of the event constituted informed consent. This process of gaining informed consent from participants aged 16 years and over was explicit in the ethics applications to the institutional committees, who thus waived the need for parental consent for participants aged 16 and 17 years old. The study was granted ethical approval by the Hull York Medical School Ethics Committee (reference number 17 26) and was granted exempt status by the Indiana University School of Medicine Institutional Review Board (reference number 1901221393).

As the audience entered the event venue, they were presented with a test sheet (S1 File) that tested their knowledge of the location of 7 different brain regions and structures, and a further question on stroke tested whether they could extrapolate from a damaged brain area to the physical consequences in the body with four options to select from. The maximum score an individual could receive was eight from eight questions. This sheet also asked for some demographic details. This was used to identify the composition of the audience attracted to the event and, for some characteristics, to explore whether this had an effect on performance in the test. Age and gender were not analyzed in the context of performance on the test, as it was not deemed appropriate to use data on protected characteristics for the purpose of this study.

After the talk and dissection, the participants completed a post-event test which asked the same questions. The pre-and post-test sheet answers keys were set up with different answer coding to ensure that anyone who simply copied between tests, rather than engaging with the activity, would be clear for data analysis and could be removed from the dataset. The pre-and post-event tests were on either side of a single sheet of paper, allowing the individual change in performance to be determined. Participation in the study aspect of the Anatomy Nights event was completely anonymous and voluntary, and completed survey sheets were given a random Participant ID to compare individuals' change in scores.

## Data analysis

We asked two general questions about the data. First, do scores differ between testing pre- vs. post- the Anatomy Nights talks and dissection? We addressed this question using a generalized linear model (R's glm function; R version 4.1.0 [21]) that assumed a binomial distribution and implemented a logit link function. Here, and throughout, we present back-transformed (generalized) means, standard errors and 95% confidence intervals (hereafter, 95% CI) for each treatment level. We present the magnitude of the effect of pre- vs. post- testing on scores as an odds ratio, (with standard error and 95% CI), calculated with R's 'emmeans' package (V 1.6.1). Throughout, odds ratio estimates (and associated error) are also based on generalized means. Throughout, 95% confidence intervals for effect sizes are not adjusted for multiple comparisons.

Second, we investigated whether the participants' Academic Qualification (school, undergraduate, postgraduate, none), employment in the healthcare sector (Yes, No), and location (Dundee, Hull, Edinburgh, Indianapolis) affected test scores. Specifically, we tested whether each of these factors affected *i)* the magnitude of the *change in score* between pre- and post-sessions (i.e., quantified by an interaction term between test timing and the focal factor), and *ii)* the average score attained. We implemented three models for each of these three factors (i.e., education, employment type, location). Model type *(a)* included a term for the test timing (pre- vs. post-), a term for the given factor (e.g., location), and a term accounting for the interaction between the given factor and timing; model type *(b)* was identical to type *(a)*, but lacked the interaction term; model type *(c)* included a term for timing, only. We compared models using likelihood ratio tests to determine whether a focal factor influenced test scores. For example, for a given factor type (e.g., employment type), we compared model type *(a)* vs. *(b)* to determine whether the interaction term affected test scores and *(b)* vs. *(c)* to test whether the focal factor affected average scores. Note that two participants had no Academic Qualification (category 'none'). We excluded these two participants from the analysis of Academic Qualification because the sample size for this group ('none') was too small to effectively compare it to the remaining three Academic Qualification categories.

We used the glmmTMB function [22] (Version 1.1.2) to model test performance with generalized mixed-effects models, implementing a binomial response distribution and logit-link

function. Models included the fixed effects terms described, above (models *a—c*), Participant ID as a random effect, and a second random effect ('units') that modeled overdispersion.

As described in Results, likelihood ratio tests revealed little evidence for interactions between test timing and each of the three focal factors (Academic Qualification, Employment type, Location). Therefore, we calculated mean test performance (averaging over test timing) and effect size for each level of each focal factor using models that lack an interaction (i.e., model type (b)). S1 Table presents these estimates and effect sizes (again) on the latent scales.

We used binomial tests to determine whether the probability of correctly answering the question regarding stroke differed significantly from the random expectation of 25%; 'pre-' and 'post-' data were analyzed separately.

We do not test whether performance changed between 'pre-' and 'post-' time periods for individual questions because exceedingly high test performance led to poor performance by Generalized Linear Models. In particular, all participants answered Question 3 correctly post-lecture and dissection (i.e., 91 / 91 answered correctly), which led to nonsensical effect size estimates (and 95% CI's) for this question. Therefore, we report results for individual questions descriptively.

All data and R scripts are available (S1 Data and S2 File) to allow readers to replicate our analyses.

## Results

### Audience demographics

All individuals attending the Anatomy Night event were invited to participate in the research. From those in attendance, a total of 102 participants sheets were collected. Nine of these were removed from the analysis because eight of them had not completed both sides of the test sheet, and one of them had copied the answers from the pre-test; two were removed due to being too young (<16 years) to meet requirements of ethical approval. The demographics of the 91 participants who completed the test sheets are shown in Table 1.

Our analyses revealed no evidence that any of the three factors (academic qualification, experience working in healthcare, Location) affected the extent to which scores changed between pre- and post-educational activity (i.e., test of interaction term; likelihood ratio tests, all *p*-values > 0.26).

Similarly, we found no evidence that academic qualification and experience working in healthcare affected the average test score (likelihood ratio tests, all *p*-values > 0.23; see Tables 2 and S1 for effect size estimates). In contrast, our models revealed strong evidence that mean test scores differed among locations (likelihood ratio test, *p* = 0.0059). Examination of effect sizes and their 95% CI's (Tables 2 and S1) suggests that Edinburgh and USA both tended to have higher scores than Dundee and Hull, but little difference occurred between Edinburgh and USA and between Dundee and Hull. Note that our models of Academic Qualification, experience working in healthcare, and Location also analyze effects of test timing on scores: these results (p-values, effect size with 95% CI's; not shown) are consistent with the results presented in "Anatomical Knowledge", below.

### Anatomical knowledge

Overall, our analyses reveal strong evidence that Anatomy Nights events increased overall test scores (Generalized linear model; z value = -9.325, *p* < 2e-16) from (generalized mean proportion ± SE) 0.639 ± 0.018 (95% CI: 0.603, 0.673) prior to the activity to 0.857 ± 0.013 (95% CI: 0.830, 0.881) (Fig 1). This result corresponds to an odds ratio (post / pre) of 3.39 ± 0.445 (95% CI: 2.62, 4.39). (These results correspond to estimated mean ± SE of (pre-)

**Table 1. Audience demographics of the Anatomy Nights brain event from 2018–19.**

| Demographic | Category | % participants ($n = 91$) |
|---|---|---|
| Location of event | Dundee, UK | 26.4 |
| | Edinburgh, UK | 26.4 |
| | Hull, UK | 22.0 |
| | Indianapolis, US | 25.3 |
| Age | 16–17 years | 2.2 |
| | 18–34 | 72.5 |
| | 35–50 | 19.8 |
| | >50 | 5.5 |
| Gender | Man | 34.1 |
| | Woman | 64.8 |
| | Gender diverse | 0 |
| | Prefer not to say | 1.1 |
| Highest educational qualification | School | 11.0 |
| | Undergraduate | 38.4 |
| | Postgraduate | 49.4 |
| | None | 1.1 |
| Works in healthcare | Yes | 28.6 |
| | No | 71.4 |

0.570 ± 0.077 (95% CI's: 0.419, 0.721) and (post-) 1.792 ± 0.106 (95% CI's 1.58, 2.00) on the logit scale, and to an estimated effect size of (post—pre) 1.22 ± 0.13 (95% CI's 0.96, 1.48) on the log odds ratio scale.)

Most questions contributed to the overall improvement between pre- and post-test scores. Fig 2, however, highlights three questions that likely contributed little to overall score improvement. Specifically, pre-test scores were very high for questions 1 and 3, leaving little scope for improvement, whereas scores improved and declined to similar degrees for question 8 (scores improved, declined, and remained unchanged for 12, 20, and 59 participants, respectively).

Question 8, regarding stroke (S1 File) had few correct answers; of 91 participants, 39 and 31 correctly answered this question at 'pre-' and 'post-' stages, respectively (Fig 2, Question 8). However, binomial tests revealed moderate to strong evidence that participants tended to correctly answer this question more often than expected due to chance (25%) at both the 'pre' ($p = 0.00022$) and 'post' ($p = 0.05238$) stages.

## Discussion

### Audience demographics

Overall, the results demonstrate the Anatomy Nights brain event increased the audience's knowledge of brain anatomy (Fig 1). The majority of the audiences were aged 18–34 (72.5%), were women (64.8%), had a university education (undergraduate 38.4%, postgraduate 49.4%), and did not work in healthcare (71.4%) (Table 1). The predominance of women and university graduates in the audience aligns with research that shows women and people with higher education are more likely to actively seek out information relating to their health [23]. Our audience proportions are also similar to those of Science Café events; however, Anatomy Nights reaches a younger age group than such events (18–34 vs. 40+) [24]. This younger demographic more closely aligns to that seen at annual Pint of Science events [25] and shows that Anatomy Nights joins a growing number of events reaching a younger audience.

**Table 2. Estimates from Generalized Linear Mixed Models for effects of academic qualfication, empoyment in healthcare, and location on test performance.** All results based on generalized means, averaged over effect of test timing (pre-, post-). 95% CI's for contrasts are not adjusted for multiple comparisons.

| Academic Qualification | | | | |
|---|---|---|---|---|
| **Level** | **Proportion answers correct** | **SE** | **95% CI** | |
| Post | 0.794 | 0.0240 | 0.743, 0.837 | |
| School | 0.797 | 0.0492 | 0.683, 0.877 | |
| Under | 0.774 | 0.0287 | 0.712, 0.825 | |
| **Contrast** | **Odds ratio** | **SE** | **df** | **95% CI** |
| post / school | 0.982 | 0.329 | 174 | 0.507, 1.90 |
| post / under | 1.127 | 0.242 | 174 | 0.737, 1.72 |
| school / under | 1.148 | 0.394 | 174 | 0.583, 2.26 |
| **Work in Healthcare** | | | | |
| **Level** | **Proportion answers correct** | **SE** | **95% CI** | |
| No | 0.773 | 0.0212 | 0.729, 0.812 | |
| Yes | 0.816 | 0.0291 | 0.752, 0.867 | |
| **Contrast** | **Odds ratio** | **SE** | **df** | **95% CI** |
| No / Yes | 0.766 | 0.172 | 177 | 0.493, 1.19 |
| **Location** | | | | |
| **Level** | **Proportion answers correct** | **SE** | **95% CI** | |
| Dundee | 0.726 | 0.0354 | 0.651, 0.790 | |
| Edinburgh | 0.837 | 0.0261 | 0.779, 0.883 | |
| Hull | 0.715 | 0.0395 | 0.631, 0.786 | |
| USA | 0.835 | 0.0272 | 0.774, 0.882 | |
| **Contrast** | **Odds ratio** | **SE** | **df** | **95% CI** |
| Dundee / Edin | 0.516 | 0.133 | 175 | 0.310, 0.859 |
| Dundee / Hull | 1.060 | 0.277 | 175 | 0.633, 1.774 |
| Dundee / USA | 0.525 | 0.137 | 175 | 0.313, 0.880 |
| Edin / Hull | 2.054 | 0.554 | 175 | 1.206, 3.499 |
| Edin / USA | 1.018 | 0.275 | 175 | 0.598, 1.733 |
| Hull / USA | 0.495 | 0.135 | 175 | 0.289, 0.849 |

As well as analysis of audience demographics, data on educational background and employment in the healthcare sector were used to ascertain if these had an effect on baseline knowledge and performance after the event. Interestingly, neither of these had a significant impact. There was, however, a significant difference between locations; reasons for this may be due to differences in the presentation styles of the hosts and/or the demographics of the audiences between locations. Regarding educational background, information on the field studied at the university level was not collected, so no conclusion on previous anatomical knowledge could be derived. However, it is notable that this group's performance after an educational activity was no different from those without a university qualification.

There was no evident difference in baseline and improvement in the group that works in the healthcare sector, which given the presence of anatomy in most health professions training, is surprising. This may be for two reasons: the vast differences in levels of anatomy education in allied health professions [26], and individuals not directly involved in patient care (e.g., hospital administrators) could reasonably have said they work in the healthcare sector. Without further detail on the attendees' role in healthcare, the effect of prior exposure to anatomical education cannot be ascertained.

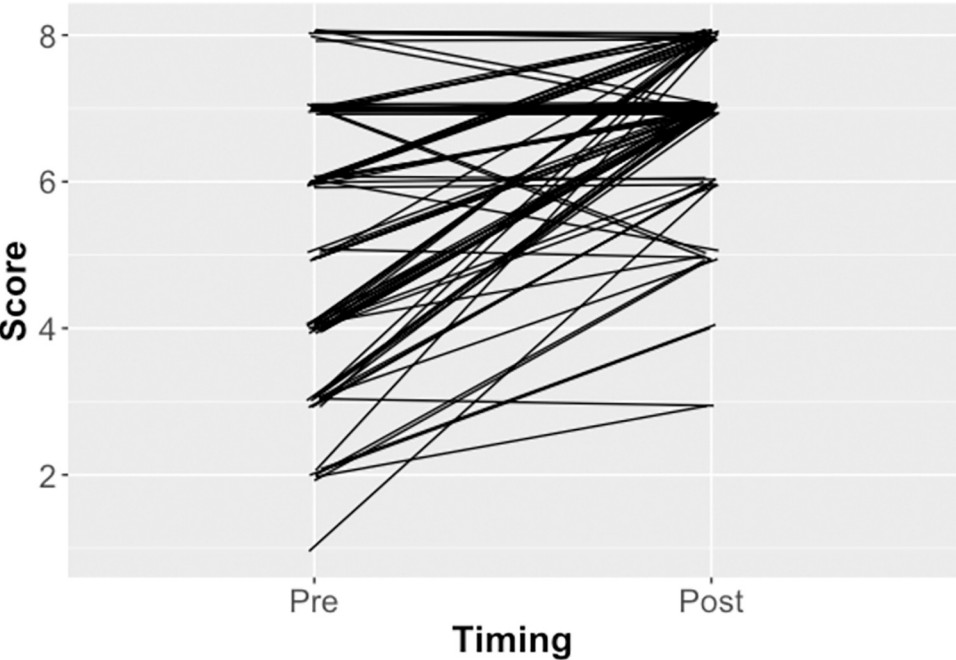

**Fig 1. Overall test scores.** Test scores obtained (out of eight) by study participants (n = 91) before ("Pre") and after ("Post") experiencing the educational activity component of Anatomy Nights. Lines connect pre- and post-test scores for individual participants.

## Public anatomical knowledge

From the results, it could be said that the general public has a reasonable baseline knowledge with regards to brain anatomy with an average score of 65% in the pre-test. However, while

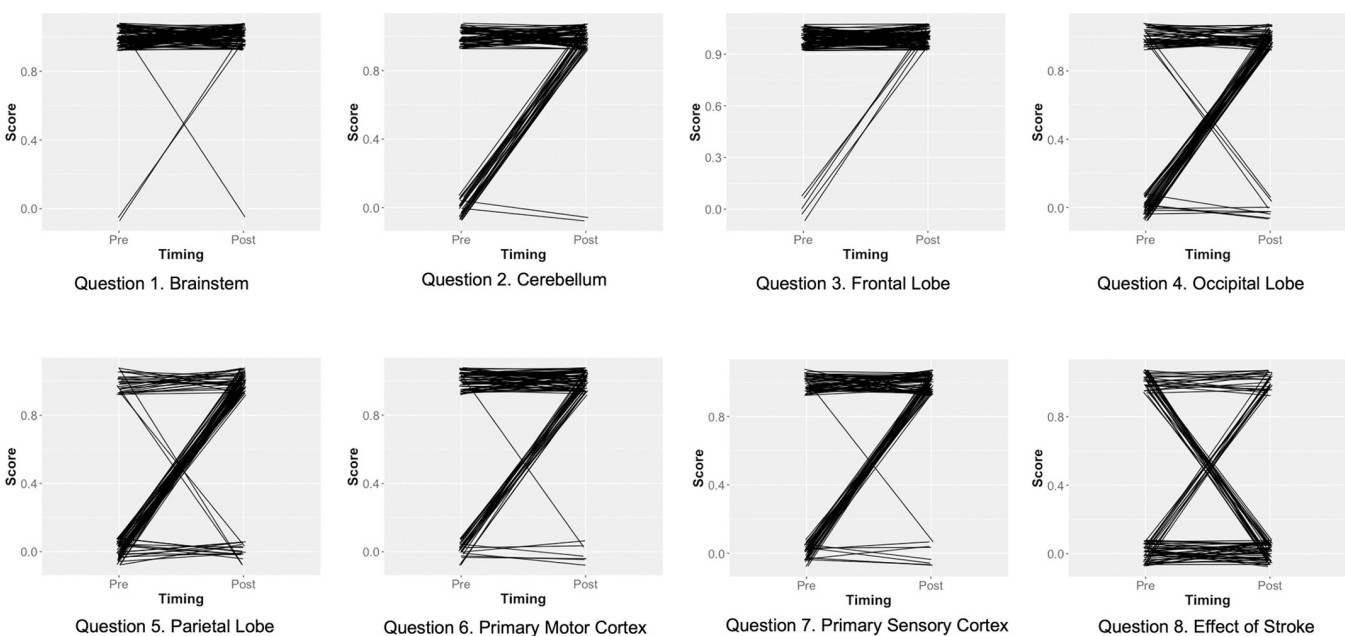

**Fig 2. Test scores for individual questions.** Test scores obtained for each question by study participants (n = 91) before ("pre") and after ("post") experiencing the educational activity component of Anatomy Nights. Lines connect pre- and post-scores for individual participants.

this is by no means a "fail" grade on the test, the performance in individual questions highlights the role that anatomical language may be playing in the differences between performance on questions. Indeed, all the questions except for 1, 3, and 8 appear to demonstrate improvement after the Anatomy Nights event (Fig 2). Questions 1 and 3 (S1 File) asked the audience to identify the "brain stem" and "frontal lobe" and had little room for improvement, with 98% and 94% answering the questions correctly in the pre-test, respectively. Compared to the other questions, 1 and 3 included words that are familiar to a non-scientific audience, namely "stem" and "frontal". The anatomical structures themselves reflect the normal definitions of these words: the brain stem descends from under the brain, like a plant's stem is under the flower; the frontal lobe is at the front of the brain. While it cannot be stated with certainty that it is the public's familiarity with the words rather than already knowing what these structures are, these were the only questions without anatomical jargon in their name. Within all science communication, jargon remains a significant barrier between scientists and the public [15], and anatomical sciences are no exception. These two questions account for 25% of the test, and if they were removed, then baseline knowledge of the general public audience would be verging on a 50% pass. It can be concluded from this that the brain stem and frontal lobe form part of the common knowledge of the general public; however, all other structures asked for in the test are not.

## Application of anatomical knowledge

Question 8 also showed no apparent change in performance between pre- and post-tests; however, this was a more complex question than the other seven. This question required the application of knowledge. An area on the brain was identified with an "X" and they were asked which area of the body would be affected by damage there. They were given four options with a combination of right or left and upper or lower limbs. To correctly answer this question, the participants needed to identify the side of the brain shown in the diagram, apply knowledge of decussation, and then overlay the map of the motor homunculus onto the diagram to determine if the upper or lower limb would be affected.

While performance on Question 8 appeared to change little between test periods (Fig 2), the audience tended to perform marginally better than would be expected by chance at both periods. It is interesting to note that the participants' responses to this question did not remain static despite similar overall performance. It can be seen in Fig 2, Question 8, that roughly similar proportions of participants either changed their answer or kept it the same between pre- and post-test, including changing from a correct answer to an incorrect one. The content of the Anatomy Nights event challenged the cognitive capacity of the audience, and Question 8 demonstrates the limits of that. The presentation of new information and the requirement to first select the appropriate content and then integrate different aspects of it to reach the correct answer may exceed a reasonable expectation of cognitive load for an audience presented with extensive volume of novel information (intrinsic load) in a short timeframe (extrinsic load) [27]. Indeed, the improvement in the overall test score demonstrates a net gain in knowledge, but most participants were unable to apply the new knowledge to different neuroanatomical contexts.

## Confounding factors

There are several factors, which could not be fully controlled for, and should be considered in the interpretation of the results presented here. While every effort was made to standardize the event across the four locations, including the provision of a template presentation, each presenter inevitably added their own style of public engagement. This is inherent in the design of

the Anatomy Nights format, and each anatomist should continue to be encouraged to develop their own style so the event can educate and develop faculty at the same time as educating the public. However, this difference in delivery could not be factored into the analysis.

Further factors, including delivery of the event in the evening, variable alcohol consumption at the venues, attendees potentially working with others or using personal smart devices to search for answers, filling in the pre-test during or after the talk and dissection, are all possible confounders and will have played some part in the performance on the pre- and post-event tests. Regardless, an increase in test performance after the event was seen and each of these factors would be considered likely to diminish this effect. Therefore, we have confidence in the positive educational influence of the Anatomy Nights event on short-term anatomical knowledge.

Finally, it is also not possible to ascertain whether the audience's new knowledge was derived from the presentation or from the dissection. It is probably a combination of the two, with the presentation being the primary source of information gain and the dissection acting as a way to consolidate this information with a 3D structure. The act of dissection is a novelty in science communication, and this distinctive way of engaging the audience will have created a learning landmark [28] for some audience members. Irrespective of whether the lecture or the dissection provided the more significant component of knowledge acquisition, the promise of the dissection component inevitably attracted some attendees and is, therefore, a crucial component of a successful Anatomy Nights event.

## Conclusion

The Anatomy Nights brain event comprising a short presentation and dissection led by an expert anatomist significantly increases the public's knowledge of the anatomy of the brain irrespective of location, educational background, and employment in the healthcare sector. However, we found no evidence that participation in Anatomy Nights improved participants' ability to apply this knowledge to neuroanatomical contexts (e.g., stroke).

Baseline anatomical knowledge most strongly aligns with structures that have a standard English name. Future events should cement this knowledge and introduce more anatomical structures to become common knowledge. A follow-up of audience members after the event could also be used to determine if acquired knowledge is retained over time.

## Supporting information

**S1 File. Pre- and post-test sheet.** The double-sided test that each participant completed before and after the event.
(PDF)

**S2 File. R Scripts from statistical analyses of the results.**
(TXT)

**S1 Table. Estimates from Generalized Linear Mixed Models for effects of academic qualification, employment in healthcare, and location on test performance.**
(DOCX)

**S1 Data. Raw data from the pre- and post-tests.**
(CSV)

## Acknowledgments

The authors would like to acknowledge all the faculty and volunteers involved in the events, the venues and management, and the attendees without whose participation, the analysis

would not have been possible. The event in Indianapolis was organized by the Science Outreach Community of Indiana University School of Medicine (SOCI) graduate student group. The event in Hull was organized by the Hull and East Yorkshire (HEY) Science group.

## Author Contributions

**Conceptualization:** Katherine A. Sanders, Janet A. C. Philp.

**Data curation:** Janet A. C. Philp, Andrew S. Cale, Claire L. Cunningham, Jason M. Organ.

**Formal analysis:** Katherine A. Sanders, Janet A. C. Philp, Crispin Y. Jordan, Jason M. Organ.

**Investigation:** Katherine A. Sanders.

**Methodology:** Katherine A. Sanders, Janet A. C. Philp, Crispin Y. Jordan.

**Project administration:** Katherine A. Sanders, Janet A. C. Philp, Andrew S. Cale, Claire L. Cunningham, Jason M. Organ.

**Writing – original draft:** Katherine A. Sanders, Janet A. C. Philp, Crispin Y. Jordan, Andrew S. Cale, Claire L. Cunningham, Jason M. Organ.

**Writing – review & editing:** Katherine A. Sanders, Janet A. C. Philp, Andrew S. Cale, Claire L. Cunningham, Jason M. Organ.

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
