## [Decision Letter · Decision Letter 0]

2 Jul 2021

PONE-D-21-16218

Anatomy Nights: An international public engagement event increases audience knowledge of brain anatomy

PLOS ONE

Dear Dr. Sanders,

Thank you for submitting your manuscript to PLOS ONE. After careful consideration, we feel that it has merit but does not fully meet PLOS ONE’s publication criteria as it currently stands. Therefore, we invite you to submit a revised version of the manuscript that addresses the points raised during the review process.

We look forward to receiving your revised manuscript.

Kind regards,

Ebrahim Shokoohi

Academic Editor

PLOS ONE

3. Please include additional information regarding the steps taken to validate the survey or questionnaire.

Furthermore, in your Methods section, please provide a justification for the sample size used in your study, including any relevant power calculations (if applicable).

Finally, please clarify any inclusion and exclusion criteria applied to the participant inclusion criteria.

4. You indicated that you had ethical approval for your study. In your Methods section, please ensure you have also stated whether you obtained consent from parents or guardians of the minors included in the study or whether the research ethics committee or IRB specifically waived the need for their consent.

Additional Editor Comments (if provided):

Dear Author

I have checked the manuscript. It contains interesting data. But the statistical analysis need to be improved by the detail information. What package you have used for the statistical analysis? Additionally, you need to check the citation for the text and the ref part to be in accordance. The rest of the comments were added to the text attached.

Regards,

Reviewers' comments:

Reviewer's Responses to Questions

**Comments to the Author**

1. Is the manuscript technically sound, and do the data support the conclusions?

Reviewer #1: Yes

2. Has the statistical analysis been performed appropriately and rigorously? 

Reviewer #1: Yes

3. Have the authors made all data underlying the findings in their manuscript fully available?

Reviewer #1: No

4. Is the manuscript presented in an intelligible fashion and written in standard English?

Reviewer #1: Yes

5. Review Comments to the Author

Reviewer #1: Abstract

No research methods, design, and data collection instrument were mentioned in the abstract

Introduction

Avoid usage of long sentences, particularly in paragraph 2. Improve referencing, for instance sentence beginning with “stroke….lacking”; requires reference. Line 75 -78

Line 90 ending with countries hosting Anatomy Nights: It will be interesting to expand the statement by pointing out that not all countries host Anatomy Nights and stating regions which commonly host such Anatomy Nights predominantly.

It will also be interesting in the background before giving the aim of the study, to outline if the Anatomy Nights are common in your study setting and the usual attendance.

Materials and methods

Outline clearly research approach and design applicable in the study. Outline target population and calculation of sample size. As alluded in the results section that participants amounted to 102, how did you arrive at it? Moreover, outline the sampling method clearly, how did sample the 102 persons. Did you sample every person attending Anatomy Nights? If so, were all in attendance over the age to consent? You also did not highlight use of informed consent. How was sample calculated, which technique was used, including how was sampling down. Did you sample every participant coming in. Include clear data analysis method used. Specify specific statistical test used.

Results

Demographic section,

Did you select every person who attended the Anatomy Nights, if so, this should be clear in the method subsection? Explain how you arrived at 102. Furthermore, as you indicated that nine participants did not fully complete the questionnaire/sheets, you should explain in your manuscript if you did consider sampling for attrition rates to cover for incomplete sheets. Although you did not. Revise accordingly.

anatomy knowledge

I suggest the inclusion of table showing pre-and-post outcomes on specific questions

Discussions and conclusion

No comments, except to say well discussed and concluded.

Overall comments

Avoid use of long sentences. Revise, proofread and edit.

Recommendation for approval

The manuscript can be revised with consideration of the comments and re-submitted for approval.

6. PLOS authors have the option to publish the peer review history of their article (what does this mean?). If published, this will include your full peer review and any attached files.

Reviewer #1: No

---

## [Author Response · Author response to Decision Letter 0]

18 Oct 2021

Reviewer #1: Abstract

No research methods, design, and data collection instrument were mentioned in the abstract

The abstract now reflects the research design better than before. We hope the reviewer finds this acceptable.

Introduction

Avoid usage of long sentences, particularly in paragraph 2. Improve referencing, for instance sentence beginning with “stroke….lacking”; requires reference. Line 75 -78

Line 90 ending with countries hosting Anatomy Nights: It will be interesting to expand the statement by pointing out that not all countries host Anatomy Nights and stating regions which commonly host such Anatomy Nights predominantly.

It will also be interesting in the background before giving the aim of the study, to outline if the Anatomy Nights are common in your study setting and the usual attendance.

Thank you for these comments, as they helped us set up our introduction to Anatomy Night much better. To be clear, these events are not common in our field and were designed precisely because they are not common and are an effort to bring anatomy education back into public spaces, consistent with the history of the discipline but not in heavy practice since the Victorian era. We have made edits to the introduction to reflect this set up better.

Materials and methods

Outline clearly research approach and design applicable in the study. Outline target population and calculation of sample size. As alluded in the results section that participants amounted to 102, how did you arrive at it? Moreover, outline the sampling method clearly, how did sample the 102 persons. Did you sample every person attending Anatomy Nights? If so, were all in attendance over the age to consent? You also did not highlight use of informed consent. How was sample calculated, which technique was used, including how was sampling down. Did you sample every participant coming in. Include clear data analysis method used. Specify specific statistical test used.

Thank you for pointing where we had unclear descriptions of the work and for identifying where we could better demonstrate effect sizes for our results. Everyone in attendance at Anatomy Night events was invited to participate in the pre- and post-event survey research. Participants were provided “participant information sheets” detailing the purpose of the quiz and how their quiz submission would be used. The inclusion of participants under the age of 16 was covered in our ethics application and approval, and it should be noted that due to the venues used for Anatomy Nights, anyone under the age of 16 would have been accompanied by an appropriate adult (also note that only 2 participants in this study were aged 16 years or under). This research, as detailed in the manuscript, was designed to assess the baseline knowledge of our participants and whether anatomical knowledge was gained from participation in Anatomy Night. We have been very explicit now about which statistical approaches we used, and how we interpret our results. Our materials and methods section has been revised to reflect this:

“We asked two general questions of the data. First, do scores differ between testing pre- vs. post- the Anatomy Nights talks and dissection? We addressed this question using a generalized linear model (R’s glm function; R version 4.1.0 [21]) that assumed a binomial distribution and implemented a logit link function. Here, and throughout, we present back-transformed (generalized) means, standard errors and 95% confidence intervals (hereafter, 95% CI) for each treatment level. We present the magnitude of the effect of pre- vs. post- testing on scores as an odds ratio, (with strandard error and 95% CI). calculated with R’s ‘emmeans’ package (V 1.6.1). Throughout, odds ratio estimates (and associated error) are also based on generalized means. 

Second, we investigated whether the participants’ Academic Qualification (school, undergraduate, postgraduate, none), employment in the healthcare sector (Yes, No), and location (Dundee, Hull, Edinburgh, Indianapolis) affected test scores. Specifically, we tested whether each of these factors affected i) the magnitude of the change in score between pre- and post- sessions (i.e., quantified by an interaction term between test timing and the focal factor), and ii) the average score attained. We implemented three models for each of these three factors (i.e., education, employment type, location). Model type (a) included a term for the test timing (pre- vs. post-), a term for the given factor (e.g., location), and a term accounting for the interaction between the given factor and timing; model type (b) was identical to type (a), but lacked the interaction term; model type (c) included a term for timing, only. We compared models using likelihood ratio tests to determine whether a focal factor influenced test scores. For example, for a given factor type (e.g., employment type), we compared model type (a) vs. (b) to determine whether the interaction term affected test scores and (b) vs. (c) to test whether the focal factor affected average scores. Note that two participants had no Academic Qualification (category ‘none’). We excluded these two participants from the analysis of Academic Qualification because the sample size for this group (‘none’) was too small to effectively compare it to the remaining three Academic Qualification categories. 

We using the glmmTMB function [22] (Version 1.1.2) to model test performance with generalized mixed-effects models, implementing a binomial response distribution and logit-link function, Models included the fixed effects terms described, above (models a - c), Participant ID as a random effect, and an second random effect (‘units’) that modeled overdispersion. 

As described in Results, likelihood ratio tests revealed little evidence for interactions between test timing and each of the three focal factors (Academic Qualification, Employment type, Location). Therefore, we calculated mean test performance (averaging over test timing) and effect size for each level of each focal factor using models that lack an interaction (i.e., model type (b)). Appendix Table A1 presents these estimates and effect sizes (again) on the latent scales.

We used binomial tests to determine whether the probability of correctly answering the question regarding stroke differed significantly from the random expectation of 25%; ‘pre-‘ and ‘post-’ data were analyzed separately.

We do not test whether performance changed between ‘pre-‘ and ‘post-‘ time periods for individual questions because exceedingly high test performance led to poor performance by Generalized Linear Models. In particular, all participants answered Question 3 correctly post-lecture and dissection (i.e., 93 / 93 answered correctly), which led to nonsensical effect size estimates (and 95% CI’s) for this question. Therefore, we report results for individual questions descriptively.

All data and R scripts are available (S2-3 Files) to allow readers to replicate our analyses.”

Results

Demographic section,

Did you select every person who attended the Anatomy Nights, if so, this should be clear in the method subsection? Explain how you arrived at 102. Furthermore, as you indicated that nine participants did not fully complete the questionnaire/sheets, you should explain in your manuscript if you did consider sampling for attrition rates to cover for incomplete sheets. Although you did not. Revise accordingly.

Our results section has been rewritten to reflect the changes described in our revised materials and methods section. See response above

anatomy knowledge

I suggest the inclusion of table showing pre-and-post outcomes on specific questions

We detail the outcomes of pre- and post-survey results for specific questions in the results and discussion sections.

Discussions and conclusion

No comments, except to say well discussed and concluded.

Thank you for the positive feedback!

Overall comments

Avoid use of long sentences. Revise, proofread and edit.

Thank you. 

Recommendation for approval

The manuscript can be revised with consideration of the comments and re-submitted for approval.

---

## [Decision Letter · Decision Letter 1]

4 Nov 2021

PONE-D-21-16218R1Anatomy Nights: An international public engagement event increases audience knowledge of brain anatomyPLOS ONE

Dear Dr. Sanders,

Thank you for submitting your manuscript to PLOS ONE. After careful consideration, we feel that it has merit but does not fully meet PLOS ONE’s publication criteria as it currently stands. Therefore, we invite you to submit a revised version of the manuscript that addresses the points raised during the review process.

We look forward to receiving your revised manuscript.

Kind regards,

Ebrahim Shokoohi

Academic Editor

PLOS ONE

Additional Editor Comments :

Dear Authors

We have received the comments from the Referees, and they have mentioned clarifying the statistical analysis. Please address the issues raised by the Referees point by point. You should describe how you did the survey based on the statistical method.

Reviewers' comments:

Reviewer's Responses to Questions

**Comments to the Author**

1. If the authors have adequately addressed your comments raised in a previous round of review and you feel that this manuscript is now acceptable for publication, you may indicate that here to bypass the “Comments to the Author” section, enter your conflict of interest statement in the “Confidential to Editor” section, and submit your "Accept" recommendation.

Reviewer #1: (No Response)

Reviewer #2: (No Response)

2. Is the manuscript technically sound, and do the data support the conclusions?

Reviewer #1: Yes

Reviewer #2: Partly

3. Has the statistical analysis been performed appropriately and rigorously? 

Reviewer #1: Yes

Reviewer #2: No

4. Have the authors made all data underlying the findings in their manuscript fully available?

Reviewer #1: Yes

Reviewer #2: Yes

5. Is the manuscript presented in an intelligible fashion and written in standard English?

Reviewer #1: Yes

Reviewer #2: Yes

6. Review Comments to the Author

Reviewer #1: The previous comments were partially addressed.

Abstract

Mention in the abstract research methods, design and data collection instrument applied in the study.

Method

How was sampling conducted?

be specific also on population, methods and design.

It would be interesting to identify the type of harm which could occur and how to measurers to address it.

Also you indicated that children under the age of 16 were accompanied by guardians or parents, but it is confusing as to whether did they consent on their behalf.The statement may need to be reworded.

Implement the major methodological recommendation for approval for publication

Reviewer #2: Dear Authors

I have checked the revised manuscript submitted. Despite the questions raised and addressed by the Authors, the statistical analysis need to be clarified in the MM. How the statistical performed?

7. PLOS authors have the option to publish the peer review history of their article (what does this mean?). If published, this will include your full peer review and any attached files.

Reviewer #1: No

Reviewer #2: No

---

## [Author Response · Author response to Decision Letter 1]

29 Nov 2021

Reviewer #1: The previous comments were partially addressed.

Abstract

Mention in the abstract research methods, design and data collection instrument applied in the study.

We have reworded to the abstract to better reflect the research methods, design, and data collection instruments in the study.

Method

How was sampling conducted? 

be specific also on population, methods and design.

Our sampling was described in the previous revision of the manuscript. However, the reviewer’s comment prompted us to re-evaluate how clear our description was. To that end, we have rearranged the first paragraph of the Data Collection section to be clearer. Our sampling strategy is clearly described in lines 148-162. Furthermore, we provide the pre- and post-test surveys as a supplemental file (S1) for complete transparency.

It would be interesting to identify the type of harm which could occur and how to measurers to address it.

Whereas the reviewer finds this to be interesting, it is not within the scope of the current research design or the scope of the manuscript. We thank the reviewer for the comment but have chosen not to address this comment because it does not work in service to the research as presented. 

Also you indicated that children under the age of 16 were accompanied by guardians or parents, but it is confusing as to whether did they consent on their behalf.The statement may need to be reworded.

As described above, we reworded this statement to clearly state that guardians/parents consented on behalf of their respective minors by submission of the survey itself. Participants were informed of this procedure for obtaining informed consent at the top of each survey page.

Implement the major methodological recommendation for approval for publication

Reviewer #2: Dear Authors

I have checked the revised manuscript submitted. Despite the questions raised and addressed by the Authors, the statistical analysis need to be clarified in the MM. How the statistical performed?

We respectfully disagree with the reviewer that the statistical analysis needs to be clarified further. In lines 186-240, we have clearly described all of the statistical tests used as well as which functions in R were used. Furthermore, as we did with the last revision, we provide supplementary files that include all the raw data from the surveys (S3), all estimates from generalized linear mixed models (S2), and the R scripts used to analyze the data (S4) so that anyone interested can re-run the analyses if they wish. This is excellent practice, exemplifies transparency, and we could not be more detailed in this manuscript.

---

## [Decision Letter · Decision Letter 2]

16 Dec 2021

PONE-D-21-16218R2Anatomy Nights: An international public engagement event increases audience knowledge of brain anatomyPLOS ONE

Dear Dr. Sanders,

Thank you for submitting your manuscript to PLOS ONE. After careful consideration, we feel that it has merit but does not fully meet PLOS ONE’s publication criteria as it currently stands. Therefore, we invite you to submit a revised version of the manuscript that addresses the points raised during the review process.

We look forward to receiving your revised manuscript.

Kind regards,

Ebrahim Shokoohi

Academic Editor

PLOS ONE

Journal Requirements:

Additional Editor Comments:

Dear Authors

I have checked the submitted revised manuscript. All the questions raised by the Referees were addressed. However, only some minor changes to the affiliation, which yellow highlighted, are needed. The text corrections also need to be considered. Please check the attached file. After the minor changes done, it can be accepted for publication.

Regards,

Ebrahim Shokoohi (Academic Editor)

Reviewers' comments:

Reviewer's Responses to Questions

**Comments to the Author**

1. If the authors have adequately addressed your comments raised in a previous round of review and you feel that this manuscript is now acceptable for publication, you may indicate that here to bypass the “Comments to the Author” section, enter your conflict of interest statement in the “Confidential to Editor” section, and submit your "Accept" recommendation.

Reviewer #1: (No Response)

Reviewer #2: All comments have been addressed

2. Is the manuscript technically sound, and do the data support the conclusions?

Reviewer #1: Yes

Reviewer #2: Yes

3. Has the statistical analysis been performed appropriately and rigorously? 

Reviewer #1: Yes

Reviewer #2: Yes

4. Have the authors made all data underlying the findings in their manuscript fully available?

Reviewer #1: Yes

Reviewer #2: Yes

5. Is the manuscript presented in an intelligible fashion and written in standard English?

Reviewer #1: Yes

Reviewer #2: Yes

6. Review Comments to the Author

Reviewer #1: Include research method in the abstract and also methodology. Be specific on population size and sample calculation. Indicate harm which could have occurred and how to minimise. The manuscript can be accepted for publication upon revision and inclusion of the critical elements pointed out

Reviewer #2: The revised paper has been improved. all comments addressed and the statistical analysis is clear as done by R

7. PLOS authors have the option to publish the peer review history of their article (what does this mean?). If published, this will include your full peer review and any attached files.

Reviewer #1: No

Reviewer #2: No

---

## [Author Response · Author response to Decision Letter 2]

3 Apr 2022

In response to the evaluation of the manuscript, we have added to the following sentence on page 7 lines 157-8:

This process of gaining informed consent from participants aged 16 years and over was explicit in the ethics applications to the institutional committees, who thus waived the need for parental concern for participants aged 16 and 17 years old.

We hope this provides the necessary clarity.

________

We have not included a response to reviewers, as the editors comments stated that only their minor edits and change to affiliations was required. We believe this is likely because the only review comment came from Reviewer 1 and we had already addressed that comment in a previous submission.

Editors comments 16/12/21

Dear Authors

I have checked the submitted revised manuscript. All the questions raised by the Referees were addressed. However, only some minor changes to the affiliation, which yellow highlighted, are needed. The text corrections also need to be considered. Please check the attached file. After the minor changes done, it can be accepted for publication.

Regards,

Ebrahim Shokoohi (Academic Editor)

---

## [Editor Report · Decision Letter 3]

12 Apr 2022

Anatomy Nights: An international public engagement event increases audience knowledge of brain anatomy

PONE-D-21-16218R3

Dear Dr. Sanders,

We’re pleased to inform you that your manuscript has been judged scientifically suitable for publication and will be formally accepted for publication once it meets all outstanding technical requirements.

Kind regards,

Ebrahim Shokoohi

Academic Editor

PLOS ONE

Additional Editor Comments (optional):

All issues and questions have been addressed.

Reviewers' comments:

no comments

---

## [Editor Report · Acceptance letter]

1 Jun 2022

PONE-D-21-16218R3 

Anatomy Nights: An international public engagement event increases audience knowledge of brain anatomy 

Dear Dr. Sanders:

I'm pleased to inform you that your manuscript has been deemed suitable for publication in PLOS ONE. Congratulations! Your manuscript is now with our production department. 

Kind regards, 

on behalf of

Dr. Ebrahim Shokoohi 

Academic Editor

PLOS ONE